# The Role of Context in Learning to Read Languages That Use Different Writing Systems and Scripts: Urdu and English

**Amna Mirza [1],\* and Alexandra Gottardo [2]**

1 Faculty of Education, Mount Saint Vincent University, 166 Bedford Highway, Halifax, NS B3M 2J6, Canada
2 Department of Psychology, Wilfrid Laurier University, Waterloo, ON N2L 3C5, Canada; agottardo@wlu.ca
\* Correspondence: amna.mirza@msvu.ca

**Abstract:** Language learning involves linguistic and societal factors that interact to facilitate or hinder second language learning. Different contextual factors provide an opportunity to examine and understand the similarities and differences that occur among bilingual children who learn the same first (L1) and second language (L2) in different countries and contexts. This paper explored the role of context, learners' profiles and linguistic differences of Urdu–English bilinguals in Canada and Pakistan. Within- and cross-linguistic comparisons were conducted for 76 Urdu–English speakers from Pakistan and 50 participants from Canada. Children, ages 8–10 years, were tested on language and literacy measures in both languages. Group comparisons of performance on language measures across languages and countries confirmed that relative strengths were based on the societal languages of each country (Urdu in Pakistan and English in Canada). Despite some similarities in relations among skills within language, differences in the language learning context provided interesting findings regarding the role of L1 skills for acquiring L2 reading skills. These findings challenge the theories developed using data from L2 learners, where learners acquire the societal language in immersion contexts, such as in North America or Europe.

**Keywords:** context-based learning; language-general mechanisms; literacy; bilingualism; linguistic skills

## 1. Introduction

Language learning involves multiple factors, including factors internal to the learner and factors related to the language being learned, as well as the learning context (see Gottardo et al. 2021 for a review). Learner factors—in this case, child factors—include memory, cognitive skills, linguistic skills and metalinguistic skills. Linguistic factors include the linguistic typology and script. Contextual factors include the instructional and societal contexts in which a language is learned, in terms of exposure to oral language and literacy. These multiple factors interact to facilitate or hinder second language learning. In the case of English, language proficiency is indispensable for the success of immigrants in English-speaking countries. English language proficiency also is considered a way to access economic and educational opportunities in many non-English-speaking countries. These different contextual factors, in terms of language and learning environments, provide an opportunity to examine and understand the similarities and differences that occur among children who learn the same first and second languages in different countries/contexts. This study examined how learner variables interact with context across different languages. Specifically, the study examined relations among language and literacy variables in Urdu–English bilinguals in Pakistan and Canada, who learn Urdu as their first language (L1) and English as their second language (L2).

### 1.1. Contextual Factors in Language Learning

Research in psycholinguistics distinguishes between the terms "second language learning/acquisition" and "foreign language learning" (Clément et al. 1977). The term

"second language acquisition" (SLA) or "second language learning"[1] is used when learning a language is essential for survival and is a societal language. For example, motivation is typically greater for a non-native learner of German residing in Germany than for a person learning German as a school subject in Canada. In the former scenario, the language is studied in a setting where it is the medium of everyday communication. The latter context involves "foreign language learning" (FLL), which occurs when a language is learned in an environment where it is not the primary vehicle for daily interaction, and input is often limited to the school setting. Because the language is not usually needed to survive, the motivation to learn the language can be different in the two language learning contexts, SLA and FLL (MacIntyre et al. 1997).

English provides a unique example in terms of the diversity of language learning contexts. English is the official and societal language of many countries. It is the official language of many countries that are not typically associated with English culture, but that were former British or American colonies (e.g., India, Pakistan, Kenya or the Philippines). Additionally, it is the *lingua franca* of many countries, where individuals learn English to improve their educational and career opportunities. Finally, worldwide, English is the most common second language and is encouraged by parents who wish to enhance their children's economic opportunities (e.g., China) (Boubtane et al. 2016; Gott and Lesgold 2000). Therefore, privileged parents in low- and middle-income countries (LMIC) choose to send their children to schools where English is taught as an additional language beginning in the elementary grades. However, the techniques used to teach languages in these countries often differ significantly from the techniques used in North America and result in different levels and skillsets being acquired in English.

In the present study, the bilingual participants[2] were learning English in two different contexts. Urdu–English bilinguals in Pakistan learn to speak and read Urdu as their first language and learned English as their second language (L2) in a FLL context. They maintain their L1 skills while acquiring their L2, showing a pattern of additive bilingualism (Grosjean 2010; Swain and Lapkin 1991). In Pakistan, children are taught to read and write English as their second or additional language, prior to learning to speak the language (Farukh and Vulchanova 2016). Urdu–English bilinguals in Canada also learn to speak Urdu as their L1 at home, but learn to speak and read English in school as their L2, where English is the language of instruction and the dominant societal language. This scenario often results in subtractive bilingualism, where students become dominant in their L2 at the cost of L1 proficiency. Therefore, many Urdu–English speaking children who begin their schooling in Canada speaking Urdu show gradual L1 loss (Mirza et al. 2017). To investigate differences across learners in these two contexts, it is important to understand how the processes of learning to read, write and speak differ in both language learning contexts. The unique contribution of this paper is that it addresses the differences among bilinguals who are learning the same first and second languages, but are in different contexts. Similarities and differences in patterns of relations among language and literacy variables within and across languages were examined across contexts, specifically in Canada and in Pakistan.

### 1.2. Second-English Language Learning in the Canadian Context

Bilingualism is common worldwide, as many people immigrate to countries to improve opportunities for themselves and their children and must learn the societal language to succeed. Reitano (2017) shows that 16.1% of the Canadian population reported being foreign born, with Urdu speakers being the fifth largest group of recent immigrants. Although Canada is officially defined as a multicultural and multilingual country, many children born to immigrant parents show a pattern of language loss, characterized by limited vocabulary and oral language proficiency in the L1 (Reitano 2017). These children begin school fluent in their L1 and learn to speak English before, or at the same time as, they learn to read. Not only is English immersion instruction conducted in the classroom, but English is also the common language of the playground, which contributes to a pattern of L1 loss and L2 dominance. Some immigrant parents in Canada attempt to preserve their L1 at home and

encourage L1 literacy through heritage language classes, which are held on the weekends. Therefore, these immigrant children can communicate orally in their L1 at various levels of proficiency and have strong oral language skills in English, often acquiring English literacy prior to literacy in their L1 (Mirza et al. 2017).

The literature on reading acquisition also suggests that the nature of reading instruction is an important variable in reading development (Al Otaiba and Fuchs 2006; Connor et al. 2004; Foorman and Torgesen 2001). Research conducted in North America found that code-based reading instruction that acknowledges the characteristics of learners was most successful (Connor et al. 2004; Foorman and Torgesen 2001). The Urdu–English bilinguals in Canada resided in Ontario, where reading instruction followed a "balanced literacy" curriculum. The curriculum included an early focus on decoding (i.e., phonemic awareness, letter-sound knowledge) and oral language, and a later focus on fluency and reading comprehension (Willows 2008).

### 1.3. Second-English Language Learning in Pakistan

In Pakistan, children learn to speak English as a foreign language after they have begun literacy instruction in English, similar to instruction in other developing countries (Asfaha et al. 2009; Dubeck et al. 2012; Farukh and Vulchanova 2016). In these countries, children are introduced to the English letter-names in their early elementary years, most often by the age of five or six. The educational system in Pakistan tends to rely on outdated teaching methods, such as the Grammar Translation Method (GTM), to teach English in elementary grade levels (Zeeshan 2013; Panezai and Channa 2017). The GTM involves translating written text from one language to another. Vocabulary and even specific sentences can be learned through rote repetition of text or orally presented words or sentences. Rote learning is defined as the memorization of information/material based on repetition (Zeeshan 2013). The goal of this method is to ensure that children quickly recall the material through frequent repetition, although any reference to meaning is incidental. Therefore, these children might not be able to use the language productively, decode novel words or understand text.

### 1.4. Linguistic Factors Related to Reading

#### 1.4.1. Specific Variables Related to Reading

The literature has reached a consensus as to the importance of several variables related to word reading. For example, there is a well-established link between phonological awareness and word reading; a language-general feature across several alphabetic languages (Georgiou et al. 2008). Phonological processing which leads to phonological recoding is considered an important element of reading because this process functions independently and allows children to recode words that they have heard but have not seen before (Ehri 1992; Share 1995). Deciphering the alphabetic principle for a given language through the appropriate mapping of graphemes to phonemes allows readers to access word-specific phonological and semantic representations (Elbro and Pallesen 2002; Perfetti 2007). Other variables related to word reading include morphological awareness (Cheung et al. 2010; Nagy et al. 2014) and vocabulary (Dixon et al. 2012; Zhang et al. 2017). Relations between these variables and word reading have been examined in monolingual English speakers, as well as within and across languages for bilinguals. Research with bilinguals seeks to identify the role of language-general and language-specific features.

#### 1.4.2. Theories of Cross-Linguistic Relations

To understand universal processes in reading, it is important to explore bilingual children's performance on typologically different languages they learn to read and the different ways that instructional methods have an impact on reading. Languages differ in terms of orthographies, and alphabetic orthographies vary in terms of how they are written. Research on reading development suggests that when children begin the process of learning to read, they must learn the code used by their language and writing system to

represent speech using "visual symbols", a language-specific feature (Ziegler and Goswami 2005). For example, alphabetic orthographies are sometimes shallow and transparent, with regular and consistent grapheme–phoneme correspondences (e.g., Spanish, Italian, voweled Arabic), or deep, with ambiguous mappings between spellings and sounds (e.g., English, French) (Bar-Kochva and Breznitz 2014). Research findings are inconclusive in terms of the relations among reading skills across languages with similar or different scripts (Gottardo et al. 2021; Shwartz et al. 2005; Saiegh-Haddad and Geva 2008). Therefore, different languages and scripts may require different or modified models of reading to explain developmental pathways and proficiencies (Nag and Snowling 2013). Learners of languages with different scripts and different linguistic typologies provide important contrasts in the search for language-specific and universal criteria for reading (Share and Daniels 2016). As a solution to inconsistencies based on reading different scripts, the psycholinguistic grain size theory was proposed by Ziegler and Goswami (2005) to explain the process of word acquisition in different languages.

The psycholinguistic grain size theory (PGST; Ziegler and Goswami 2005) explains word reading acquisition for different alphabetic languages, which is relevant to the current study. The PGST includes three factors that contribute to the process of reading acquisition: availability, consistency and granularity. Availability refers to the ease of access of different sound units in speech. Consistency can be seen in the associations between each sound and symbol of the language. Granularity refers to the optimal level of mappings between the sounds and symbols in a language in terms of larger or smaller units. Therefore, the PGST model of reading acquisition explains the process of reading development as the abstraction of optimal mappings between orthographic units and sounds of the language that differ according to writing systems (Branum-Martin et al. 2012).

The languages in the current study, English and Urdu, are alphabetic languages, but are written with different scripts. English is written from left to right using the Roman alphabet. Urdu (L1) is written from right to left in Nastaliq script, a modified Arabic/Persian script, which can be written with or without vowel diacritics. When vowel diacritics are included, the orthography/phonology relationship of Urdu is shallow or transparent, with letters having consistent letter–sound mappings and each letter producing one sound in a word.

*1.5. Cross-Linguistic Relationships among Languages and Literacy Skills*

Research has examined cross-linguistic relationships among language and literacy skills of bilingual and biliterate people in L2 learning environments (August and Shanahan 2006; Gottardo et al. 2021; Prevoo et al. 2016). Research on L2 learning has documented the nature of relationships across languages with similar alphabetic scripts (e.g., Spanish-English) (Lindsey et al. 2003; Proctor et al. 2006) and different alphabetic scripts (e.g., Russian-English and Hebrew-English) (Abu-Rabia 2001; Wade-Woolley and Geva 2000). Progress has also been made in research comparing alphabetic and non-alphabetic languages (e.g., Chinese-English) (Chow et al. 2005; Gottardo et al. 2006; Pasquarella et al. 2015). In most cases, these bilingual immigrants learn to speak a language before they learn to read or sometimes learn to speak and read simultaneously (Bialystok et al. 2005; Snow et al. 1998).

These studies suggest that language and literacy skills are related to each other and L1 and L2 skills can influence each other (Chang 2013; Jiang 2004; Koda 1996; Melby-Lervåg and Lervåg 2011). Relationships across languages are described in the linguistic interdependence hypothesis and the script dependent hypothesis. The linguistic interdependence hypothesis states that strong L1 skills are related to strong L2 skills (James Cummins 1979), as shown by cross-linguistic relationships within the constructs, which are consistent with the language-general features of two languages (e.g., morphological skills and phonological awareness) (James Cummins 1979). Other researchers have suggested that some linguistic skills are more likely to be related across languages (Durgunoğlu 2002; Geva and Wang 2001). The script dependent hypothesis highlights the role of differences in scripts, with cross-linguistic relations being greater for languages with similar orthographies than for

languages with different orthographies, a language-specific feature supporting transfer across two different languages (Geva and Siegel 2000). Many studies that have informed the development of theories of L2 reading acquisition have been conducted in North America or Europe, where L2 learners are acquiring the majority language, often English (Share 2008). In cases where languages share an alphabetic script or a common learning environment, it is more difficult to determine which variables are related to reading skill across languages (Lindsey et al. 2003) or geographic locations (Bialystok et al. 2005). Therefore, researchers also should compare learners of typologically different languages to better understand these relationships across languages (Nag and Snowling 2013).

1.5.1. Research Questions of the Present Study

1.  Are L1 (Urdu) measures related to L2 (English) measures for phonological and morphological awareness across typologically different orthographies?
2.  What are the patterns of cross-linguistic relationships based on the similarities or differences in orthographies or linguistic typologies?
3.  What are the unique predictors of English and Urdu word reading within and across the languages in different learning contexts?

1.5.2. The Present Study

The present study explored the role of language-general and language-specific mechanisms that influence the relations across typologically different orthographies, specifically Urdu and English. Additionally, the role of context in relation to learning to read among Urdu–English bilinguals in Pakistan and Canada was explored. Within- and cross-linguistic relationships among variables in both languages were examined across countries. It was expected that context would be an important variable driving group differences in language acquisition. As described earlier, Urdu–English bilinguals in Pakistan learn to speak Urdu (L1) in their home and acquire Urdu literacy upon starting their school. Urdu remains their dominant and societal language. English literacy instruction often starts in the early elementary years at or around 5 years of age, and occurs prior to learning to speak English. Conversely, in Canada, Urdu–English bilingual children learn to speak Urdu and English simultaneously in their home. These children begin English literacy instruction prior to Urdu literacy, which in the case of these participants, is acquired at weekend language schools. For these children, English is the societal language and becomes their dominant language. Therefore, these highlighted differences in language learning are labeled "context" and used to determine the order of variables in regression analyses. Data on the populations included in this study are limited. Other comparisons of these samples learning Urdu and English in the two different contexts of this study do not exist, to the best of our knowledge. Therefore, our approach was exploratory.

## 2. Materials and Methods

### 2.1. Design

Within- and cross-linguistic comparisons were conducted among Urdu–English bilinguals from Pakistan and Canada. Groups were created based on participants' place of residence, either Pakistan or Canada.

### 2.2. Participants

*Total sample:* The participants, 126 eight-to-ten-year-old children, were tested in their country of residence. The sample included 76 Urdu–English bilinguals in Pakistan, 30 boys and 46 girls ($M_{age}$ = 9.02, *SD* = 0.88), and 50 Urdu–English bilinguals in Canada, 22 boys and 28 girls ($M_{age}$ = 8.88, *SD* = 0.82).

*Canadian Sample:* Fifty Canadian Urdu–English bilingual children who were in grades 3 to 5 in their public schools were recruited for this study. English is the societal and dominant language in the part of Canada where the families resided; therefore, the children's education was predominantly in English. These children were also registered in weekend

international language classes offered by the local school board. Children's length of enrollment in the weekend language schools varied from a minimum of 6 to a maximum of 24 months. Parents of these children had immigrated to Canada from Pakistan and usually had a background of Urdu as their national language, which was also their dominant and first language. However, in some cases, they also learned other regional languages, which they speak in their home, with Urdu being their L2. Families from Pakistan often choose to send their children to weekend language schools to maintain their Urdu language skills and their connections with their heritage and their relatives in Pakistan. At the weekend language school, the medium of instruction is Urdu, and teachers are fluent in Urdu. Teachers provide instruction on both Urdu oral language and literacy skills. Children attend this weekend language school for 5 h on Saturdays from October to May of each school year.

*Pakistani Sample:* Children recruited from Pakistan were enrolled in grades 3 to 5 in public schools, where they learned both Urdu and English literacy, simultaneously. The children were not registered in any additional weekend language school classes or tutoring. In Pakistan, public schools are further divided into Urdu medium and English medium of instruction. This particular sample was recruited from English medium public schools to meet a level of English proficiency somewhat comparable to the Canadian sample. English medium public schools in Pakistan teach both Urdu (L1) and English (L2) simultaneously. Children start schooling as fluent Urdu speakers, but learn to read Urdu in school. However, English literacy is taught in schools prior to learning to speak the language. Children sometimes start schooling with some familiarity with English words (acquired through digital media), but no formal literacy instruction.

### 2.3. Demographic Measures

Demographic information was collected through a questionnaire completed by the parents of each participant. This questionnaire was designed to identify the percentage of language use and the child home literacy environment. The questionnaire also addressed parental linguistic abilities in English and their education level (see Table 1). Parental years of formal education/qualifications was compared for both groups in Pakistan ($M = 14.70$, $SD = 0.952$) and in Canada ($M = 15.56$, $SD = 0.837$). An independent samples *t*-test revealed significant differences ($t_{(124)} = -5.19$, $p < 0.001$) between the groups of parents. Parents of Urdu–English bilingual children in Canada had higher levels of education/qualifications. This higher level of education is not surprising, as education level is one of the conditions to apply for and receive access to skills-based Canadian immigration status (Reitano 2017). These differences were also consistent when parents were asked to report their linguistic abilities in English (see Table 1). Although these group comparisons showed significant differences between parental education and linguistic abilities, teaching approaches used in schools and environmental and societal differences between both countries were considered as more important factors in describing differences across both samples. Therefore, parental education was not used as a variable of interest in any of the further analyses.

**Table 1.** Summary of demographic information for Urdu–English bilinguals in Pakistan and Canada.

| Demographics | % of Participants in Pakistan | % of Participants in Canada |
|---|---|---|
| Language used in home | 56% Urdu, 31% Punjabi, 13% other regional languages | 87% Urdu, 13% other regional languages |
| Child's best language at home | 97% used Urdu, 3% did not answer | 31% used Urdu, 64% English, 5% did not answer |
| Child's frequency of watching TV in L1 | 87% in English | 96.8% in English |
| Book reading | 91% in both Urdu and English | 86% only in English |
| Parental linguistic abilities in English | 56% fluent speakers and readers | 92% fluent speakers and readers |
| Parental education (Fathers) | 34% masters, 54% undergraduates | 78% masters, 13% undergraduates, 9% did not answer |
| Parental education (Mothers) | 86% undergraduates | 94% masters, 6% undergraduates |

*2.4. English Measures*

A battery of standardized English measures was administered to each participant in Pakistan and Canada. The measures assessed the following areas: word and pseudo-word reading, vocabulary knowledge, phonological awareness, orthographic knowledge and morphological skills. All the English measures were standardized tests that exhibited high reliability and validity based on the manual. Since norms for each standardized measure were developed for children in English contexts, for both groups of participants tested in the study, a Cronbach's alpha was calculated to measure the internal consistency of each measure. Although reliabilities were lower for the sample in Pakistan for the English tests, adaptations to standardized tests are not recommended and might have decreased the reliabilities for the sample in Canada.

*English word reading*. Two subtests of the Woodcock Reading Mastery Test, Word Identification and Word Attack (Woodcock 1998) were used to measure English word and pseudoword reading ability. Words ranged from high-frequency monosyllabic words to lower-frequency multisyllabic words. Cronbach's alpha for the sample tested in Canada was $\alpha = 0.89$ for Word Identification and $\alpha = 0.87$ for Word Attack. Cronbach's alpha for the sample tested in Pakistan was 0.67 for Word Identification and 0.71 for Word Attack.

*English vocabulary knowledge* was assessed by a measure of expressive vocabulary, the Expressive One Word Picture Vocabulary Test (EOWPVT-SBE, Brownell 2000). This test measured the ability to name pictures of objects, actions and concepts. Cronbach's alpha for the sample tested in Pakistan was 0.76 and for the sample tested in Canada was 0.84.

*English Phonological Processing Skills* were measured using the Elision subtest of the Comprehensive Test of Phonological Processing (CTOPP; Wagner et al. 1999). The Elision subtest involves repeating an orally presented word and then deleting specified phonemes. Cronbach's alpha for the sample tested in Pakistan was 0.79 and for the sample tested in Canada was 0.81.

*English Orthographic Skills* were measured by two subtests of one widely used experimental task (adapted from Olson et al. 1994). Each subtest had fifteen items with two practice items each. One subtest required participants to select the correct spelling of real words. Each item presented two possible spellings of the target word, and both spellings were phonetically correct (e.g., dream vs. dreem). The other subtest of the orthographic skills measure consisted of made-up words with two possible spellings of the words. However, one spelling was orthographically plausible in English (e.g., ploin vs. ployn; correct response: ploin). A total score of 30 could be achieved on this measure. The calculated Cronbach's alpha was 0.87 for each subsection for the sample tested in Canada. Cronbach's alpha for the sample tested in Pakistan was 0.61.

*English Morphological Skills* were measured with a morphological decomposition task that consisted of 28 items, each with 2 practice items at the beginning of the task. Children were given a multi-morphemic word, and they had to remove the morpheme to create the appropriate word (i.e., the given word was "driver" and the sentence was "Children are too young to"; The answer in this case was "drive"). The calculated Cronbach's alpha was 0.71 for the sample tested in Canada. Cronbach's alpha for the sample tested in Pakistan was 0.68.

*2.5. Urdu Measures*

Urdu–English bilinguals from both countries were assessed on Urdu word and pseudo-word reading, vocabulary knowledge, phonological awareness and morphological skills. To account for the Urdu literacy level of the children in Canada who had not yet attended weekend language school for long enough to learn the orthographic rules in their L1, children from both countries were not tested on an Urdu orthographic choice task. Additionally, no standardized measures were available in Urdu; therefore, most of the measures were translated and adapted from the English into Urdu by the primary investigator of this study. Some tasks, such as the measure of phonological awareness and morphological decomposition, could not be directly translated. Therefore, they were adapted from En-

glish into Urdu by the primary investigator to reflect the phonological and morphological features of Urdu. Since no standardized Urdu measures and norms were available, it was not possible to report the reliability from the published manuals; therefore, the Cronbach's alpha was calculated and reported for each Urdu measure based on each subsample of participants.

*Urdu word reading.* Because Urdu standardized measures of word reading were not available, the primary investigator created two wordlists by taking words from children's Urdu textbooks based on the grade 3 and 4 curriculum in Pakistan. Item lists were created, with and without vowels, and consisted of 30 items in each list. The Cronbach's alpha was calculated separately for each list by country for each task. The Cronbach's alpha was 0.92 for word reading with vowels and 0.75 for word reading without vowels for the Urdu–English bilinguals in Pakistan, and 0.83 for word reading with vowels and 0.89 for word reading without vowels for the Urdu–English bilinguals in Canada.

*Urdu vocabulary* was measured using a translated version of the Expressive One Word Picture Vocabulary Test (EOWPVT-SBE, Brownell 2000). Since this measure of expressive vocabulary was a translated version of the English vocabulary test, participants were tested on the Urdu vocabulary test on a separate day and were encouraged to complete all items. However, when they began to reach the equivalent of the ceiling (making errors or responding "Don't know."), they were shown six pictures on a page and were asked if they know the names of the pictures. They were given five seconds to decide whether they knew the name of any of the pictures, before they moved to the next set of pictures. This procedure was used to avoid frustration with this task. The Cronbach's alpha was 0.94 on this measure for the Urdu–English bilinguals in Pakistan and 0.91 for the Urdu–English bilinguals in Canada.

*Urdu phonological processing* was measured by creating a parallel version of the Elision task in Urdu, similar to the Comprehensive Test of Phonological Processing (CTOPP; Wagner et al. 1999). Children had to delete one phoneme from a real Urdu word to create another real word. Because this measure was not a standardized measure of phonological awareness in Urdu, no stopping rule was used. The calculated Cronbach's alpha was 0.63 for the Urdu–English bilinguals in Pakistan and 0.90 for the Urdu–English bilinguals in Canada.

*Urdu morphological processing* task was created by the primary investigator to reflect the morphological properties of Urdu. There were ten items in this measure. Each item consisted of a real root word. Children were asked to provide at least three derived words that could be created based on the given root word. The Cronbach's alpha was 0.97 on this measure for Urdu–English bilinguals in Pakistan and 0.78 for Urdu–English bilinguals in Canada.

## 3. Results

*Descriptive Statistics and Group Comparisons (Pakistan and Canada)*

All 76 participants in Pakistan and 50 participants in Canada were included in the analyses. Table 2 displays the means and standard deviations for each task for the participants, separated by group. Visual inspection of the data showed no floor or ceiling effects for any of the Urdu and most of the English measures, except for scores on English orthographic processing for the Canadian group, which was close to the ceiling. This suggests that non-significant relations could be a result of a restricted range and that future versions of the task should be more difficult to increase the range of potential responses.

An independent samples t-test revealed no gender differences in terms of the performance of the participants from both countries on any of the Urdu and English measures; $p$-values ranged from $p = 0.797$ to $0.652$. Therefore, gender was not included as a variable of interest in any further analyses.

**Table 2.** Mean comparisons among Urdu–English bilinguals across countries (Pakistan and Canada).

| Construct & (No of Items) | Mean (SD) Pakistan, N = 76 | Mean (SD) Canada, N = 76 | T-Value & Sig. |
|---|---|---|---|
| Urdu measures | | | |
| Word reading with vowels (30) | 22.51 (6.35) | 11.14 (2.30) | −12.12 *** |
| Word reading without vowels (30) | 19.71 (4.79) | 9.34 (1.93) | −14.53 *** |
| Vocabulary (170) | 47.23 (12.87) | 20.52 (5.00) | −13.96 *** |
| Phonological awareness task (10) | 7.07 (1.94) | 5.22 (1.05) | −6.12 *** |
| Morphological awareness task (10) | 6.07 (1.42) | 1.78 (1.14) | −17.78 *** |
| English measures | | | |
| Word reading (106) | 67.57 (11.05) | 72.68 (9.86) | 2.75 *** |
| Pseudo-word reading (45) | 23.28 (5.60) | 28.72 (5.44) | 5.39 *** |
| Vocabulary (170) | 54.80 (21.83) | 81.16 (15.59) | 7.35 *** |
| Phonological awareness task (20) | 10.68 (3.06) | 13.24 (4.53) | 3.77 *** |
| Morphological task (28) | 16.80 (3.40) | 13.16 (5.38) | −4.65 *** |
| Orthographic choice task (30) | 22.46 (2.49) | 26.36 (3.51) | 7.82 *** |

*Note: p* value < 0.001 = ***, *p* value < 0.01 = **, *p* value < 0.05 = *.

Comparisons of the English and Urdu measures were made across the groups of Urdu–English bilinguals, based on the country of residence, using independent sample t-tests. Given the number of comparisons, a conservative *p*-value of less than 0.01 was considered significant. As expected, significant differences were found for all of the measures. Not surprisingly, Urdu–English bilinguals from Pakistan had higher scores on the Urdu measures as compared to Urdu–English bilinguals in Canada. Urdu–English bilinguals from Canada had higher scores on the English measures as compared to Urdu–English bilinguals from Pakistan (see Table 2).

RQ1: Are L1 (Urdu) measures related to L2 (English) measures across typologically different orthographies?

A parallel set of correlational analyses was conducted for each bilingual group to answer this research question. Significant relationships were reported for Urdu–English bilinguals in Pakistan (see Table 3 for details) for most of the variables across both languages. English word and pseudo-word reading, vocabulary knowledge and phonological awareness were strongly to moderately negatively correlated with most of the Urdu variables tested in the study (see Table 3). On the other hand, Urdu and English measures for the participants in Canada were moderately positively correlated within constructs for word reading, vocabulary and phonological and morphological awareness (see Table 4 for details).

**Table 3.** Cross-linguistic (Urdu with English) relationships for Urdu–English bilinguals in Pakistan.

| Columns—Urdu | 1. Words | 2. Words without Vowels | 3. Vocab | 4. Phonological Awareness | 5. Morphological Decomposition |
|---|---|---|---|---|---|
| Rows—English | | | | | |
| 1. Words | −0.735 ** | −0.559 ** | −0.656 ** | −0.687 ** | −0.328 ** |
| 2. Pseudo−words | −0.588 ** | −0.449 ** | −0.510 ** | −0.488 ** | −0.234 * |
| 3. Vocab | −0.799 ** | −0.602 ** | −0.690 ** | −0.729 ** | −0.338 ** |
| 4. Phonological Awareness | −0.509 ** | −0.427 ** | −0.402 ** | −0.390 ** | −0.218 |
| 5. Morphological Decomposition | 0.706 ** | −0.626 ** | −0.615 ** | −0.648 ** | −0.316 ** |
| 6. Orthographic Choice | −0.135 | −0.313 ** | −0.397 ** | −0.165 | 0.160 |

Note: All the rows include variables in English (L2), and columns include variables in Urdu (L1). *p* value < 0.001 = ***, *p* value < 0.01 = **, *p* value < 0.05 = *.

**Table 4.** Cross-linguistic (Urdu with English) relationships for Urdu–English bilinguals in Canada.

| Columns—Urdu | 1. Words | 2. Words without Vowels | 3. Vocab | 4. Phonological Awareness | 5. Morphological Decomposition |
|---|---|---|---|---|---|
| Rows—English | | | | | |
| 1. Words | 0.303 * | 0.146 | 0.437 ** | 0.544 ** | 0.305 * |
| 2. Pseudo-words | 0.003 | −0.063 | 0.321 * | 0.316 * | 0.068 |
| 3. Vocab | 0.263 | 0.323 * | 0.429 ** | 0.548 ** | 0.360 * |
| 4. Phonological Awareness | 0.194 | 0.244 | 0.581 ** | 0.517 ** | 0.371 ** |
| 5. Morphological Decomposition | 0.283 * | 0.357 * | 0.540 ** | 0.507 ** | 0.530 ** |
| 6. Orthographic Choice | −0.019 | −0.018 | 0.327 * | 0.193 | 0.005 |

Note: All the rows include variables in English (L2), and columns include variables in Urdu (L1). *p* value < 0.001 = ***, *p* value < 0.01 = **, *p* value < 0.05 = *.

RQ2: What are the patterns of cross-linguistic relationships based on the similarities or differences in orthographies or linguistic typologies?

This research question investigated whether L1 (Urdu) phonological and morphological skills predict L2 (English) reading or reading-related skills across both bilingual groups in Pakistan and Canada. This research question was addressed by conducting two parallel, multivariate, general linear model analyses (see Tables 5 and 6). Urdu phonological awareness and morphological skills were entered as two independent (predictor) variables, and English reading skills, phonological awareness and vocabulary were used as outcome variables. The first multivariate general linear model analysis conducted on Urdu–English bilinguals from Pakistan revealed the main effect of Urdu phonological awareness as a significant predictor of all English reading (WID ($F_{(6,44)}$ = 15.29, $\eta p^2$ = 0.676, $p < 0.001$), WAT ($F_{(6,44)}$ = 6.57, $\eta p^2$ = 0.473, $p < 0.001$)) and reading-related variables (vocabulary ($F_{(6,44)}$ = 18.00, $\eta p^2$ = 0.711, $p < 0.001$) and phonological awareness ($F_{(6,44)}$ = 6.37, $\eta p^2$ = 0.465, $p < 0.001$)) (see Table 5 for details). The second language-general mechanism (morphological awareness) seemed not to be a predictor of any of the L2 variables. Similarly, the interaction between Urdu phonology and morphology was not a significant predictor of the English variables.

**Table 5.** Patterns of cross linguistic relationships for Urdu–English bilinguals in Pakistan.

| Constructs | $R^2$ | Df | F | Sig. |
|---|---|---|---|---|
| Urdu Phonological Awareness (main effect) | 0.575 | 6.44 | | |
| English Word Reading (WID) | | | 150.29 | 0.000 |
| English Non-word Reading (WAT) | | | 60.57 | 0.000 |
| English Vocabulary (EOWPVT) | | | 180.00 | 0.000 |
| English Phonological Awareness (CTOPP) | | | 60.37 | 0.000 |
| Urdu Morphological Awareness (main effect) | 0.789 | 6.44 | | |
| English Word Reading (WID) | | | 0.160 | 0.986 |
| English Non-word Reading (WAT) | | | 0.636 | 0.701 |
| English Vocabulary (EOWPVT) | | | 0.331 | 0.917 |
| English Phonological Awareness (CTOPP) | | | 10.00 | 0.434 |
| Phonological Awareness X Morphological Awareness (interaction) | 0.581 | 16.44 | | |
| English Word Reading (WID) | | | 0.727 | 0.752 |
| English Non-word Reading (WAT) | | | 0.493 | 0.938 |
| English Vocabulary (EOWPVT) | | | 0.509 | 0.929 |
| English Phonological Awareness (CTOPP) | | | 0.491 | 0.938 |

Note: *p* value < 0.001 = ***, *p* value < 0.01 = **, *p* value < 0.05 = *.

**Table 6.** Patterns of cross linguistic relationships for Urdu–English bilinguals in Canada.

| Constructs | $R^2$ | Df | F | Sig. |
|---|---|---|---|---|
| Urdu Phonological Awareness (main effect) | 0.426 | 4.35 | | |
| English Word Reading (WID) | | | 10.96 | 0.122 |
| English Non-word Reading (WAT) | | | 10.67 | 0.179 |
| English Vocabulary (EOWPVT) | | | 20.27 | 0.081 |
| English Phonological Awareness (CTOPP) | | | 30.36 | 0.020 * |
| Urdu Morphological Awareness (main effect) | 0.431 | 3.35 | | |
| English Word Reading (WID) | | | 10.26 | 0.300 |
| English Non-word Reading (WAT) | | | 10.49 | 0.232 |
| English Vocabulary (EOWPVT) | | | 0.306 | 0.821 |
| English Phonological Awareness (CTOPP) | | | 10.25 | 0.304 |
| Phonological Awareness X Morphological Awareness (interaction) | 0.470 | 7.35 | | |
| English Word Reading (WID) | | | 10.34 | 0.257 |
| English Non-word Reading (WAT) | | | 20.21 | 0.056 |
| English Vocabulary (EOWPVT) | | | 0.549 | 0.791 |
| English Phonological Awareness (CTOPP) | | | 0.542 | 0.796 |

*Note: p* value < 0.001 = \*\*\*, *p* value < 0.01 = \*\*, *p* value < 0.05 = \*.

The second parallel, multivariate, general linear model analysis conducted on Urdu–English bilinguals from Canada revealed only one main effect, specifically, Urdu phonological awareness predicted a significant change in English phonological awareness ($F_{(4,35)} = 3.36$, $\eta p^2 = 0.278$, $p = 0.020$). The remaining analyses showed no significant main effects or interactions (see Table 6 for details).

RQ3: What are the unique predictors of English and Urdu word reading within and across languages in different learning contexts?

The last research question of this study was answered by conducting parallel within and cross-linguistic hierarchical regressions analyses for each bilingual group separately. Analysis (multivariate general linear model analysis) conducted for the previous research question informed the construction of the subsequent regression model that provided us with the unique variance given by each variable in Urdu and English word reading.

Variables related to reading in Urdu–English bilinguals.

To further explore unique variance among within- and cross-linguistic variables related to Urdu and English word reading for each group, hierarchical regression analyses were conducted. The analyses were conducted separately for each group because of the different pattern of correlations. Un-voweled word reading was chosen as the Urdu word reading measure, given that it is a more authentic reading task in Urdu. Variables were included in the regression analyses based on significant correlations and theoretical considerations, which were also supported by the context in which children were learning these languages.

Within-language unique predictors of Urdu word reading. Children in Pakistan were taught to read both languages using a whole word memorization technique, specifically, rote repetition. Therefore, no formal explicit instruction on phonological or phonemic awareness was provided for the children. As a result, morphological awareness was entered as the first predictor variable in the hierarchical regression analysis. This variable captures children's knowledge of whole words and larger meaning units (root and bound morphemes), as well as the order of skills being taught at school. Phonological awareness and vocabulary were the other two variables entered in the hierarchical analysis. For the participants in Pakistan, the total variance explained for Urdu word reading was $R^2 = 0.572$, $F_{(3, 70)} = 31.12$, $p < 0.001$ (see Table 7). Urdu phonological awareness was the only variable uniquely related to Urdu word reading ($\beta = 0.530$, $t_{(70)} = 3.45$, $p = 0.001$), while vocabulary approached significance ($\beta = 0.274$, $t_{(70)} = 7.93$, $p = 0.058$).

**Table 7.** Within-language predictors of Urdu word reading for (1) participants in Pakistan, (2) participants in Canada.

| Step—Variables | ΔR² | β for Step & Sig. | Final β | Final *t*-Value & Sig. |
|---|---|---|---|---|
| **Participants in Pakistan** | | | | |
| 1. Morphological Awareness | 0.112 | 0.334 ** | −0.036 | −0.397 |
| 2. Phonological Awareness | 0.437 | 0.768 *** | 0.530 | 30.45 ** |
| 3. Vocabulary | 0.023 | 0.274 *** | 0.274 | 10.93 |
| **Participants in Canada** | | | | |
| 1. Morphological Awareness | 0.498 | 0.705 *** | 0.673 | 50.50 *** |
| 2. Phonological Awareness | 0.003 | 0.069 | 0.080 | 0.593 |
| 3. Vocabulary | 0.001 | −0.022 | −0.022 | −0.180 |

*Note: p* value < 0.001 = ***, *p* value < 0.01 = **, *p* value < 0.05 = *.

The same analysis was conducted for the participants in Canada. The total variance explained for un-voweled Urdu word reading was $R^2 = 0.502$, $F_{(3,46)} = 15.43$, $p < 0.001$ (see Table 7). Urdu morphological awareness was the only variable that was a significant unique predictor of Urdu word reading ($β = 0.673$, $t_{(46)} = 5.50$, $p < 0.001$).

Within-language unique predictors of English word reading. Based on the relationships explained in the correlational analyses and theory about predictors of word reading, we conducted a hierarchical regression analysis to explore significant predictors of English word reading. English orthographic choice, morphology, phonological awareness and vocabulary were added to the model in each step. For the participants in Pakistan, the total variance explained for English word reading was $R^2 = 0.784$, $F_{(4,69)} = 62.74$, $p < 0.001$ (see Table 8). English vocabulary was the only variable uniquely related to English word reading ($β = 0.719$, $t_{(69)} = 7.37$, $p < 0.001$), although other variables were related in prior steps. For the participants in Canada, the same analyses yielded a total variance explained for English word reading of $R^2 = 0.322$, $F_{(4,45)} = 5.33$, $p = 0.001$ (see Table 8). English vocabulary was the only variable that was a unique significant predictor of English word reading ($β = 0.417$, $t_{(45)} = 2.46$, $p = 0.018$), although other variables were significant in prior steps.

**Table 8.** Within-language predictors of English word reading for (1) participants in Pakistan, (2) participants in Canada.

| Step—Variables | ΔR² | β for Step & Sig. | Final β | Final *t*-Value & Sig. |
|---|---|---|---|---|
| **Participants in Pakistan** | | | | |
| 1. Orthographic Choice | 0.164 | 0.406 *** | 0.074 | 10.13 |
| 2. Morphological Awareness | 0.423 | 0.743 *** | 0.153 | 10.55 |
| 3. Phonological Awareness | 0.027 | 0.211 * | 0.014 | 0.187 |
| 4. Vocabulary | 0.102 | 0.719 *** | 0.719 | 70.37 *** |
| **Participants in Canada** | | | | |
| 1. Orthographic Choice | 0.140 | 0.295 * | 0.164 | 10.16 |
| 2. Morphological Awareness | 0.047 | 0.229 | 0.021 | 0.134 |
| 3. Phonological Awareness | 0.043 | 0.270 | 0.067 | 0.373 |
| 4. Vocabulary | 0.092 | 0.417 * | 0.417 | 20.46 * |

*Note: p* value < 0.001 = ***, *p* value < 0.01 = **, *p* value < 0.05 = *.

Cross-linguistic unique predictors: English variables related to Urdu word reading. Due to the exploratory nature of the study, the order of entry for the variables in hierarchical regression analyses was determined by the strength of the relationship found in the correlational analyses. To explore cross-linguistic predictors of Urdu word reading, English orthographic choice, phonological and morphological awareness and vocabulary were entered in each step of the hierarchical regression analyses. For the participants in

Pakistan, the total variance explained for Urdu word reading was $R^2 = 0.445$, $F_{(4,69)} = 13.80$, $p < 0.001$ (see Table 9). English morphological awareness and vocabulary were the only variables uniquely related to Urdu word reading ($\beta = -0.321$, $t_{(69)} = -2.02$, $p = 0.047$ and $\beta = -0.318$, $t_{(69)} = -2.03$, $p = 0.046$, respectively). For the participants in Canada, the total variance explained for Urdu word reading was $R^2 = 0.208$, $F_{(4,45)} = 2.94$, $p = 0.030$ (see Table 9). English morphological awareness was the only variable uniquely related to Urdu word reading ($\beta = 0.385$, $t_{(45)} = 2.08$, $p = 0.043$). For the sample in Canada, relationships were positive, while for the sample in Pakistan, cross-linguistic relationships were negative.

**Table 9.** English variables related to Urdu word reading among Urdu–English bilinguals in (1) Pakistan and (2) Canada.

| Step—Variables | $\Delta R^2$ | $\beta$ for Step & Sig. | Final $\beta$ | Final $t$-Value & Sig. |
|---|---|---|---|---|
| Participants in Pakistan | | | | |
| 1. Orthographic Choice | 0.175 | −0.419 *** | −0.168 | −10.61 |
| 2. Phonological Awareness | 0.082 | −0.308 ** | 0.044 | 0.359 |
| 3. Morphological Awareness | 0.154 | −0.530 *** | −0.321 | −20.02 * |
| 4. Vocabulary | 0.034 | −0.318 * | −0.318 | −20.03 * |
| Participants in Canada | | | | |
| 1. Orthographic Choice | 0.002 | −0.042 | −0.262 | −10.69 |
| 2. Phonological Awareness | 0.089 | 0.338 * | −0.093 | −0.372 |
| 3. Morphological Awareness | 0.095 | 0.424 * | 0.385 | 20.08 * |
| 4. Vocabulary | 0.022 | 0.253 | 0.253 | 10.09 |

*Note: p* value < 0.001 = ***, *p* value < 0.01 = **, *p* value < 0.05 = *.

Cross-linguistic unique predictors: Urdu variables related to English word reading. To explore cross-linguistic predictors of English word reading, Urdu variables were entered in a stepwise hierarchical analysis in the following steps: Urdu phonological and morphological awareness, and vocabulary. For the participants in Pakistan, the total variance explained for English word reading was $R^2 = 0.490$, $F_{(3,70)} = 22.43$, $p < 0.001$ (see Table 10). Urdu phonological awareness was the only variable uniquely related to English word reading ($\beta = -0.467$, $t_{(70)} = -2.79$, $p = 0.007$). However, this relationship was negative. For the participants in Canada, the total variance explained for English word reading was $R^2 = 0.326$, $F_{(3,46)} = 7.40$, $p < 0.001$ (see Table 10). Urdu phonological awareness was uniquely positively related to English word reading ($\beta = 0.430$, $t_{(46)} = 2.75$, $p = 0.008$).

**Table 10.** Urdu variables related to English word reading among Urdu–English bilinguals in (1) Pakistan and (2) Canada.

| Step—Variables | $\Delta R^2$ | $\beta$ for Step & Sig. | Final $\beta$ | Final $t$-Value & Sig. |
|---|---|---|---|---|
| Participants in Pakistan | | | | |
| 1. Phonological Awareness | 0.462 | −0.680 *** | −0.467 | −20.79 ** |
| 2. Morphological Awareness | 0.004 | 0.065 | 0.045 | 0.448 |
| 3. Vocabulary | 0.024 | −0.284 | −0.284 | −10.83 |
| Participants in Canada | | | | |
| 1. Phonological Awareness | 0.296 | 0.544 *** | 0.430 | 20.75 ** |
| 2. Morphological Awareness | 0.001 | 0.036 | 0.010 | 0.071 |
| 3. Vocabulary | 0.029 | 0.203 | 0.203 | 10.40 |

*Note: p* value < 0.001 = ***, *p* value < 0.01 = **, *p* value < 0.05 = *.

## 4. Discussion

This study compared the Urdu and English language and reading skills of Urdu–English bilinguals in Pakistan and Canada. For both groups, the variables selected for the analyses accounted for a relatively large amount of variance in the dependent variables. However, these Urdu–English bilinguals differed from each other in terms of their language

learning contexts, based on their country of residence. For the Urdu–English bilingual group from Pakistan, Urdu is the societal and home language. The children learned to speak their first language (Urdu) at home and began to read Urdu at school at the age of five or six. These bilinguals were taught to read English, their L2, prior to learning to speak English. Additionally, English exposure and instruction occurred almost exclusively in a school setting. In Pakistan, English was taught using rote memorization, while links to the underlying structure of the word (phonology, morphology) and the meaning of the word occurred incidentally, if at all. In contrast, Urdu–English bilinguals in Canada learned to speak Urdu and English simultaneously in their homes. They learned to speak English, the societal language, through interactions with peers and adults, prior to English literacy instruction. The Canadian participants usually learned to read English prior to learning to read Urdu in their schools at around the age of five.

These two linguistic and instructional contexts resulted in different patterns of cross-linguistic relationships. For the group in Pakistan, cross-linguistic analyses revealed negative relations across English and Urdu measures. The negative correlations show that for this group, strong L1 skills did not facilitate, and possibly might inhibit, L2 skills, contrary to patterns commonly found for bilingual learners (Jim Cummins 1991; Durgunoğlu 2002). It is possible that this pattern of negative relationships is related to the type of instruction or the context of English use. These children were taught English and Urdu language and literacy skills by the "whole word memorization technique", with no English and Urdu instruction on phonological and morphological awareness or any explicit focus on word meaning. These children were not provided any instruction on letter–sound knowledge, segmenting or blending as part of their primary literacy instruction. They also had fewer opportunities to practice English skills outside the classroom. Comparisons of the performance across groups confirm contextual differences, where both groups had different patterns of results on the morphological awareness test.

Although the children in Pakistan did not seem to use L1 skills to facilitate L2 learning, positive relations were found for some of the within-language variables. English vocabulary knowledge was positively related to English word reading performance for this group, suggesting that the meaning and print forms of words were related within language. These relations could be a result of the reading skills facilitating incidental vocabulary learning in English, or conversely, vocabulary assisting in partial phonological recoding. Alternately, both vocabulary and word reading might be related to general L2 proficiency in this group.

On the other hand, Urdu–English bilinguals from Canada learn to read English through classroom instruction that includes teaching of letter–sound correspondences, as well as phoneme segmentation and blending. Analysis of skills across languages showed positive relations among vocabulary, phonemic and morphological awareness for the Canadian group. This Urdu–English bilingual group learned to speak their L1, Urdu, and L2, English, simultaneously at home. Therefore, the positive relationships between Urdu and English vocabulary and measures of phonology and morphology are consistent with research that shows a relationship between L1 and L2 skills, with strong L1 skills facilitating L2 skills through a language-general mechanism (Jim Cummins 1991). These differing cross-linguistic patterns based on the language learning context provide insight into the importance of considering the instructional and societal contexts and suggest that language-general mechanisms are dependent on the language learning contexts. These differing findings across groups showed that oral language use plays an important role in L2 acquisition (Chang 2013; Jiang 2004; Koda 2008; Prevoo et al. 2016).

Group comparisons of the performance of Urdu–English bilinguals on both Urdu and English measures across countries, Pakistan and Canada, confirmed that relative strengths were based on the societal language of each country. However, the groups did not differ on the English measure of word reading. Scores on the standardized and experimental measures and responses regarding language use provided by the parents suggest that for the Canadian group, English, their L2, was their dominant language. Canadian Urdu–English bilinguals were able to read Urdu words as a result of instruction in written Urdu

that occurred once a week at weekend language schools. On the other hand, Urdu–English bilinguals in Pakistan showed a pattern of L1 dominance. They started learning to read English simultaneously with learning to read Urdu language at approximately five or six years old. However, their English "reading" skills involved memorizing written forms of words, likely with little word analysis or comprehension. Researchers found similar results in studies conducted with bilinguals in similar educational contexts in LMIC, where the children are taught to recognize written words in English, their L2, prior to learning to speak the language (Asfaha et al. 2009; Dubeck et al. 2012).

Within-language comparisons revealed significant relationships among variables across both groups. English vocabulary was a consistent unique predictor of English word reading across groups. Studies have shown inconclusive results regarding the relationship between vocabulary and word reading in English L2 learners. Some studies showed no significant relationship between the two variables (e.g., Geva et al. 2000; Gottardo et al. 2001), while others found a small, but significant relationship (Gottardo 2002; Páez and Rinaldi 2006). Although the groups in this study did not differ on English word reading, differences in vocabulary suggest that different mechanisms might be responsible for the relationships between word reading and vocabulary. The level of proficiency in English might explain why vocabulary was a predictor in some studies, but not in others (Geva 2006). Geva (2006) suggests that vocabulary may be a proxy for general L2 proficiency, with a greater impact on word reading being found for more proficient L2 learners (Geva 2006). This explanation is potentially valid for the participants in Canada. However, this explanation is less likely for the participants in Pakistan, given that their vocabulary skills are relatively lower than their word reading skills. Alternately, vocabulary might assist both groups of English learners in determining the pronunciation of unfamiliar written words with irregular letter–sound patterns, in cases where letter-by-letter decoding is less effective, such as in the word "yacht" (Gottardo et al. 2017; Kirby et al. 2008; Share 1995; Tunmer and Chapman 2012).

Although the learner experiences and linguistic context differed for the participants, the languages being learned were held constant. This approach differs from previous research, where English as L2 learners with different L1s are compared in terms of English skills and relations between languages (Bialystok et al. 2005; Gottardo et al. 2016; O'Brien et al. 2019; Pasquarella et al. 2015). These studies and similar studies conducted with immigrant children examined language acquisition in contexts where the L2 (often English) is the majority societal language (August and Shanahan 2006). This body of research has been influential in the development of theories of language and literacy acquisition, such as the linguistic interdependence hypothesis, which shows cross-linguistic relationships among language and literacy skills of bilingual and biliterate learners (Jim Cummins 1991). Specifically, research conducted in North America and Europe shows that L1 language and literacy skills are related to L2 skills (see August and Shanahan 2006). These relations between skills exist when both languages that the bilinguals learn are alphabetic (Abu-Rabia 2001; Lindsey et al. 2003; Mirza et al. 2017). However, theories of L2 reading acquisition have not been tested in English as a foreign language context, specifically, across a wide range of languages with large differences in typology and/or scripts.

Finally, to conclude, the multivariate linear model analysis from RQ2 determined that a main effect of Urdu phonological awareness was significantly related to English measures for the bilingual children in Pakistan. The Canadian sample did not reveal similar relationships, as only Urdu and English phonological awareness were related to each other. There were no other significant main effects or interactions revealed for either of the groups. Furthermore, the hierarchical regression analyses from RQ3 provided us with unique variance given by each variable tested in the model/s across both countries' bilingual groups, which revealed the unique and most interesting findings of the current study, namely, the role of language skills in relation to reading across contexts. Although learning languages outside of the dominant societal context has been examined, it is often in the context of two languages sharing the Roman alphabet, such as English and Spanish

(Sparks et al. 2017) or English and French (Mady 2013). However, the role of the language learning context has not been explicitly examined in relation to these hypotheses in previous studies. The data from the current study showed negative correlations for the participants from Pakistan, suggesting inhibition between the Urdu and English skills, which was not consistent with Jim Cummins (1991). However, the positive correlations between the Urdu and English variables for the group in Canada were consistent with Jim Cummins (1991). Therefore, future research studies and theories should consider the linguistic and instructional context in which the participants learn a language when examining predictors of reading. The findings of this study also suggest that current theories of L1 to L2 relations formulated based on research with bilinguals in the North American and European contexts should be tested and potentially adapted for the large number of L2 learners who learn English in a foreign language setting with different instructional techniques (Share 2008). As English is increasingly considered a global language, understanding the factors related to acquisition of English language and literacy skills in other learning contexts is important.

The present findings examine the role of L1 skills in relation to L2 word reading. The linguistic interdependence hypothesis (Jim Cummins 1991) is known as a "universal hypothesis", which addresses the underlying cognitive processes that contribute to the literacy development in different languages, regardless of orthography. In contrast, the script dependent hypothesis states that variations in relations across orthographies can influence the ease of reading acquisition across multiple languages (Geva and Siegel 2000; Frost et al. 1987). Additionally, the psycholinguistic grain size theory suggests that different processes are used to read different languages. Finally, the negative relationship across languages for the participants in Pakistan raises the question of what it means to "know" a language, considering the different instructional approaches practiced across cultures and countries. Further research should be conducted examining languages written in scripts other than the Roman alphabet.

The design of the study is a strength and a weakness. The different language learning contexts contributed to the unique findings, which can be attributed to differences in participants' schooling experiences. However, both the educational and linguistic/societal contexts differed for the two groups, which made it more difficult to interpret the findings. Finally, the lower reliabilities for the English tasks for the group in Pakistan likely resulted from the different instructional and linguistic contexts experienced by this group and the fact that the tests were normed on native speakers of English.

## 5. Conclusions

In conclusion, despite some similarities in within-language relations, differences in the instructional and linguistic context provided novel findings regarding the role of L1 skills in L2 reading acquisition. Overall, the findings suggest that children tested in Pakistan had higher skills in Urdu phonological awareness, but had lower scores on all English measures used in the study. On the other hand, in the Canadian sample, there was a positive relationship among all the measures across both languages. These findings challenge the theories developed using data from L2 learners, for whom language learning occurs in an immersion context, where learners are acquiring the societal language, as is common in the North American or European context. Specifically, the applicability of linguistic interdependence should be examined and possibly adapted to other linguistic contexts, including foreign language learning. Overall, research conducted with bilingual children across languages and cultures enhances our understanding of the processes of language and literacy teaching and learning and the challenges that learners face in L2 acquisition.

**Author Contributions:** Conceptualization, A.M.; methodology, A.M.; software, A.M.; validation, A.M., A.G.; formal analysis, A.M.; investigation, A.M.; resources, A.M., A.G.; data curation, A.M., A.G.; writing—original draft preparation, A.M.; writing—review and editing, A.G.; visualization, A.M.; supervision, A.G.; project administration, A.M. All authors have read and agreed to the published version of the manuscript.

**Funding:** This research project received no external funding.

**Institutional Review Board Statement:** The study was conducted in accordance with an approval given by the Institutional Review Board (or Ethics Committee) of Wilfrid Laurier University (protocol code 4779 and date of approval: 29 January 2016).

**Informed Consent Statement:** Informed consent was obtained from all subjects involved in the study.

**Data Availability Statement:** Data is unavailable due to privacy or ethical restrictions.

**Conflicts of Interest:** The authors declare no conflict of interest.

## Notes

[1]  Second language learning and second language acquisition are differentiated to refer to learning a language through instruction and incidental learning, respectively. In the case of the participants in Canada, English learning occurs through both mechanisms.

[2]  For the purposes of this study, bilingualism is broadly defined as "knowing" two languages.

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
