# Peer review of "The Role of Context in Learning to Read Languages That Use Different Writing Systems and Scripts: Urdu and English"

_languages, doi:10.3390/languages8010086_

Round 1

Reviewer 1 Report

Overall Comments.

Thank you for the opportunity to review the manuscript entitled: “The Role of Context in Learning to Read Languages that use 2 Different Writing Systems and Scripts: Urdu and English”. I appreciated the clarity throughout the introduction. The literature review was clear and justified the research design very well. I also think the nuance of variables (i.e., language with different scripts and different language learning contexts) are extremely important to investigate in terms of understanding potential differences in literacy development. Nonetheless, given the complexity of all the conditions, I did find it difficult to follow the analysis plan and understand the information provided in the tables. Below you will see some specific comments. I think the work is very valuable and would be in favour of publication, if my suggestions/questions are addressed.  

Suggestions:

p. 6, line 267. The description of the school experience for the Canadian sample is comprehensive. More detail could be provided about the Pakistani sample. For example, how many hours per week are these students typically learning each language? Does it begin upon entering school?

p. 6. Line 282-286. Although I am fine with you not using parental variables in your analyses, could you please be more specific about what “prominent enough” means with respect to the children’s variables? 

Materials section:  I appreciate the difficulty of assessing language skills when there are no standardized tests in a particular language, and I commend the researchers on their efforts here. I also appreciated that reliability measures were reported within the samples for Urdu measures. I would recommend doing the same thing for the English measures. The fact that these bilingual students are not really the population on which the English measures were normed suggests to me that running Cronbach’s Alpha for each of these samples would be worthwhile.

Results section & Tables.

Here I found the accessibility of the findings to be challenging both with respect to the text and the tables. For each of the tables, can you be more specific about what values/analyses are being presented? Here are my concerns about each table and its corresponding analyses (where applicable).

Table 1: it is overly complicated and does not consistently include information. For example, it reports B.A. level in one set of participant but not the other. It is also not clear what is meant by “fluent”. Small point, the centre justification of the text is distracting.

Table 2: Could you use identing more effectively so that you don’t have to keep repeating the language (see example)? Also, there are only 50 students in Canada and the “*” are not telling the reader the significance level.  

Urdu

       XXXXX

       XXX

English

       XXXX

       XXXX

For Table 3a and Table 3b, I struggled to understand what analyses were undertaken here. There is mention of relationships, so I assumed that you had done regressions (as you have said). However, the language surrounding “main effects”, then led me to believe you were doing some sort of ANOVA. But the independent variables were continuous. Either way, I did not understand how many analyses were done, and what the results in Table 3 were telling us. If you have one predictor variable and one outcome variable, my understanding is that you are just doing a correlation (even if you set up a regression). Consider being careful about how you are presenting the research questions because it was not clear to me how your first question led to this analysis.

For Tables 4a and 4b, could you provide information on what the acronyms mean? There should also be Urdu listed in the table itself (but there is no need to repeat English for each measure in the first column). I was confused about why the correlations were presented after the information from table 3.

For Table 5 through 8, Could you make it explicit that below the line is the second set of participants. I also struggled here to understand the input methodology. Were each variable entered individually into the model? Would the results have differed if you had but them all in at the same time? If so, is there a need to emphasize the significance level at the step where the variable was inputted? I also wondered if chronological age was considered as a control variable, given the age spread in your sample.

Discussion

For the discussion, I can appreciate that there were quite a few different patterns of results, and it was difficult to delve into a discussion of each regression analysis. Given the diverse pattern of results, how confident would you be about replication here?

Author Response

Reviewer 1 (Thank you so much for your review. My responses to your comments and suggestions are addressed in red).

Overall Comments.

Thank you for the opportunity to review the manuscript entitled: “The Role of Context in Learning to Read Languages that use 2 Different Writing Systems and Scripts: Urdu and English”. I appreciated the clarity throughout the introduction. The literature review was clear and justified the research design very well. I also think the nuance of variables (i.e., language with different scripts and different language learning contexts) are extremely important to investigate in terms of understanding potential differences in literacy development. Nonetheless, given the complexity of all the conditions, I did find it difficult to follow the analysis plan and understand the information provided in the tables. Below you will see some specific comments. I think the work is very valuable and would be in favour of publication, if my suggestions/questions are addressed.  

Suggestions:

  1. 6, line 267. The description of the school experience for the Canadian sample is comprehensive. More detail could be provided about the Pakistani sample. For example, how many hours per week are these students typically learning each language? Does it begin upon entering school? Information regarding schools in Pakistan has been added in text on p 6.
  2. 6. Line 282-286. Although I am fine with you not using parental variables in your analyses, could you please be more specific about what “prominent enough” means with respect to the children’s variables? “prominent enough” is replaced with “held important position in describing differences across both samples” to make it sound clear.

Materials section:  I appreciate the difficulty of assessing language skills when there are no standardized tests in a particular language, and I commend the researchers on their efforts here. I also appreciated that reliability measures were reported within the samples for Urdu measures. I would recommend doing the same thing for the English measures. The fact that these bilingual students are not really the population on which the English measures were normed suggests to me that running Cronbach’s Alpha for each of these samples would be worthwhile. Calculated and added in the paper.

Results section & Tables.

Here I found the accessibility of the findings to be challenging both with respect to the text and the tables. For each of the tables, can you be more specific about what values/analyses are being presented? Here are my concerns about each table and its corresponding analyses (where applicable).

Table 1: it is overly complicated and does not consistently include information. For example, it reports B.A. level in one set of participant but not the other. It is also not clear what is meant by “fluent”. Small point, the centre justification of the text is distracting. All of these comments and suggestions have been addressed in Table 1.

Table 2: Could you use identing more effectively so that you don’t have to keep repeating the language (see example)? Also, there are only 50 students in Canada and the “*” are not telling the reader the significance level.  This information has been clearly added to all the tables.

Urdu

       XXXXX

       XXX

English

       XXXX

       XXXX

For Table 3a and Table 3b, I struggled to understand what analyses were undertaken here. There is mention of relationships, so I assumed that you had done regressions (as you have said). However, the language surrounding “main effects”, then led me to believe you were doing some sort of ANOVA. But the independent variables were continuous. Either way, I did not understand how many analyses were done, and what the results in Table 3 were telling us. If you have one predictor variable and one outcome variable, my understanding is that you are just doing a correlation (even if you set up a regression). Consider being careful about how you are presenting the research questions because it was not clear to me how your first question led to this analysis. The details were given in result section. Please see p 9, line 402 – 419 for the details of analyses provided in table 3a and 3b.

“This research question investigated whether L1 (Urdu) phonological and morphological skills predict L2, English, reading or reading related skills across both bilingual groups in Pakistan and Canada. This research question was addressed by conducting two parallel multivariate regression analyses. Urdu phonological awareness and morphological skills were entered as two independent (predictor) variables and English reading skills, phonological awareness and vocabulary were used as outcome variables. The first multivariate analysis conducted on Urdu-English bilinguals from Pakistan revealed the main effect of Urdu phonological awareness as a significant predictor of all English reading (WID, F (6,44) = 15.29, p < .001, WAT (F (6,44) = 6.57, P < .001) and reading related variables (vocabulary (F (6,44) = 18.00, p < .001 and phonological awareness (F (6,44) = 6.37, p < .001) (see Table 3a for details). The second language-general mechanism (morphological awareness) seemed not to be a predictor of any of the L2 variables. Similar was the case for the interaction of both predictor variables.

            The second parallel multivariate analysis conducted on Urdu-English bilinguals from Canada revealed only one main effect, specifically Urdu phonological awareness predicted a significant change on English phonological awareness, F (4,35) = 3.36, P = .020.  The remaining analyses showed no significant main effects or interactions (see Table 3b for details)”.

For Tables 4a and 4b, could you provide information on what the acronyms mean? There should also be Urdu listed in the table itself (but there is no need to repeat English for each measure in the first column). I was confused about why the correlations were presented after the information from table 3. Table 4a and 4b presents correlational analyses different from what is presented in table 3a and 3b. details of the analyses were provided on p 9. Line 422 – 430.

“A parallel set of correlational analyses was conducted across both bilingual groups to answer this research question. Significant relationships were reported for Urdu-English bilinguals in Pakistan (see Table 4a for details) for most of the variables across both languages. English word and pseudo-word reading, vocabulary knowledge and phonological awareness were strongly to moderately negatively correlated with most of the Urdu variables tested in the study (See Table 4a). On the other hand, Urdu and English measures for the participants in Canada were moderately positively correlated within constructs for word reading, vocabulary and phonological and morphological awareness (see Table 4b for details)”.

For Table 5 through 8, Could you make it explicit that below the line is the second set of participants. Addressed.

I also struggled here to understand the input methodology. Were each variable entered individually into the model? Yes. Please see details mentioned on p. 9 line 437 to 496. Would the results have differed if you had but them all in at the same time? Explained the reason why not “line 437 – 496”. If so, is there a need to emphasize the significance level at the step where the variable was inputted? I also wondered if chronological age was considered as a control variable, given the age spread in your sample. Given the circumstances children attend language school in Canada, chronological age does not contribute as a confounding variable as much as the time in language school. Sometimes, children who are 6 years of age learn same content as children who are 8 or 9 years old. Their grade level and content are determined by their literacy level.

Discussion

For the discussion, I can appreciate that there were quite a few different patterns of results, and it was difficult to delve into a discussion of each regression analysis. Given the diverse pattern of results, how confident would you be about replication here? Parts of these results are replication of a study conducted on a different language sample (manuscript in progress) by the same group of researchers. Therefore, I am confident to state that these results are replicable in future studies conducted on similar linguistic sample (demographically). (addressed on line 628 – 629).

Reviewer 2 Report

General comments:  A strength of this paper is the data collected in Pakistan with Urdu-English bilingual children. It is exciting to see more linguistic work examining languages with different scripts and non-Western contexts. The authors have interesting questions related to the context of learning to read in languages with different writing systems and scripts, both related to societal context and instructional context. Unfortunately, it is not clear that the design of the study and the differing populations allow them to answer those questions. Below I first address the most serious concern I have, namely the between-group comparison.  I think provide a list of other questions and concerns about measures and terminology.

Main Concern: Between-Group comparison Issue:  I can think of several reasons that comparing the 76 Urdu-English speakers in Pakistan aren’t comparable with the 50 participants in Canada.  First, and foremost , is the issue of language dominance.  The 8-10 year olds in Canada would much more be classified as English dominant heritage speakers of Urdu, whereas the children in Pakistan are clearly Urdu dominant second language learners of English.  Both the societal presence of English and the nature of language instruction differs between the two groups, making it difficult to indicate which factor plays a more salient role. The educational backgrounds of the parents also differs significantly between the two groups.  Answering RQ3 “What is the role of the learning context in determining relationships across languages?” is not possible given these confounding factors.  Ideally, the authors would have used a language dominance measures (like the Bilingual Language Profile Birdsong et al.) and then criterion matched participants across the two groups based on dominance, parental education, age, etc. This would have made a stronger comparison of the impact of instructional methods.  Although I think the authors should abandon a direct between-group comparison, I think the can reframe the paper around RQs 1 and 2 and testing cross-linguistic relationships in languages with different scripts.

Additional comments:

Pages 5-6:  On these pages the authors mention theories of cross-linguistic relations and relationships between languages and literacy skills.  RQ1 examines the linguistic interdependence hypothesis and the script dependent hypothesis. Is RQ2 meant to test the PGST?  Why is PGST discussed?  This is unclear.

Page 6:  The author state that morphological skills and phonological awareness are ‘general features’ of two languages (based on Cummins).  What is the theory behind selecting these two features as ‘general features’?  I think unpacking that relationship a bit for your reader and relating it to you results would be useful.

Page 6:  What do you mean by language-specific mechanisms?  It would be helpful to your reader to explicitly state what your language general and language specific mechanisms are.  What is the theoretical underpinnings of this generality and specificity? 

Page 8:  The fact that some of the children in Canada were not literate in Urdu reinforces my concern above that they are heritage learners, and not comparable to the learners in Pakistan.

Page 9:  The two groups, which did differ on every other measure, didn’t differ on English word reading. Why?  On its face, this seems strange and merits a discussion.  The word reading test was made up of two subtests, word identification and word attack.  Did you do a global scoring for word reading?  Were the two groups different on each measure?  Was it something about word vs. pseudoword reading?  I think there needs to be some level of convincing the reader that this different on this measure is indeed explainable given the differences on all of the other measures.  Is it meaningful in some way?

Page 13-14:  Given my concerns about comparing the two groups, I am not comfortable with the claims about the effects of instructional methods and the societal languages. The authors hint at dominance issues and then on page 14 also indicate the conflation of educational and linguistic/societal conflation.  I like the story, truly I do, but the empirical evidence is not strong enough to make those claims.

Author Response

Reviewer 2: (Thank you so much for your review. My responses to your comments and suggestions are addressed in red).

General comments:  A strength of this paper is the data collected in Pakistan with Urdu-English bilingual children. It is exciting to see more linguistic work examining languages with different scripts and non-Western contexts. The authors have interesting questions related to the context of learning to read in languages with different writing systems and scripts, both related to societal context and instructional context. Unfortunately, it is not clear that the design of the study and the differing populations allow them to answer those questions. Below I first address the most serious concern I have, namely the between-group comparison.  I think provide a list of other questions and concerns about measures and terminology.

Main Concern: Between-Group comparison Issue:  I can think of several reasons that comparing the 76 Urdu-English speakers in Pakistan aren’t comparable with the 50 participants in Canada.  First, and foremost , is the issue of language dominance.  The 8-10 year olds in Canada would much more be classified as English dominant heritage speakers of Urdu, whereas the children in Pakistan are clearly Urdu dominant second language learners of English.  Both the societal presence of English and the nature of language instruction differs between the two groups, making it difficult to indicate which factor plays a more salient role. The educational backgrounds of the parents also differs significantly between the two groups.  Answering RQ3 “What is the role of the learning context in determining relationships across languages?” is not possible given these confounding factors.  Ideally, the authors would have used a language dominance measures (like the Bilingual Language Profile Birdsong et al.) and then criterion matched participants across the two groups based on dominance, parental education, age, etc. This would have made a stronger comparison of the impact of instructional methods.  Although I think the authors should abandon a direct between-group comparison, I think they can reframe the paper around RQs 1 and 2 and testing cross-linguistic relationships in languages with different scripts. We expected these differences and used these as a confirmatory analyses in RQ3.

Additional comments:

Pages 5-6:  On these pages the authors mention theories of cross-linguistic relations and relationships between languages and literacy skills.  RQ1 examines the linguistic interdependence hypothesis and the script dependent hypothesis. Is RQ2 meant to test the PGST?  Why is PGST discussed?  This is unclear. No, RQ 2 is not meant to test PGST. PGST is included in the paper to explain the process of reading acquisition among bilinguals. Since, Cummin’s theory “linguistic interdependence hypothesis” discusses languages and their relationships. Linguistic interdependence hypothesis does not address general features across languages but only language specifics. Reasoning behind the mentioning of PGST has been added in the paper on p. 4, line 165 – 167.

Page 6:  The author state that morphological skills and phonological awareness are ‘general features’ of two languages (based on Cummins).  What is the theory behind selecting these two features as ‘general features’?  I think unpacking that relationship a bit for your reader and relating it to you results would be useful. Cummins does not talk about specific constructs in his theories and models of learning to read but discusses just the language specifics. General features across languages are missing in his theories. This explanation has been added in the paper. See p.4, line 165 - 167.

Page 6:  What do you mean by language-specific mechanisms?  It would be helpful to your reader to explicitly state what your language general and language specific mechanisms are.  What is the theoretical underpinnings of this generality and specificity? Please see explanation on language-specific and language-general mechanisms on p. 5, line 201 – 220.

Page 8:  The fact that some of the children in Canada were not literate in Urdu reinforces my concern above that they are heritage learners, and not comparable to the learners in Pakistan. We expected these to be different learners but they do speak Urdu. They are born in Canada, they speak Urdu at home, they also have exposure of English at home, they speak Urdu fluently as their L1.

Page 9:  The two groups, which did differ on every other measure, didn’t differ on English word reading. Why?  The t-value on word reading was a significant t-value with p less than .01. Asterisk has been added, which was a typo in previous version. On its face, this seems strange and merits a discussion.  The word reading test was made up of two subtests, word identification and word attack.  Did you do a global scoring for word reading?  Were the two groups different on each measure?  Was it something about word vs. pseudoword reading?  I think there needs to be some level of convincing the reader that this different on this measure is indeed explainable given the differences on all of the other measures.  Is it meaningful in some way?

Page 13-14:  Given my concerns about comparing the two groups, I am not comfortable with the claims about the effects of instructional methods and the societal languages. The authors hint at dominance issues and then on page 14 also indicate the conflation of educational and linguistic/societal conflation.  I like the story, truly I do, but the empirical evidence is not strong enough to make those claims. It’s difficult to create completely parallel measures which will allow us to compare across language within each subject. A decision was made to not conduct within subject comparisons.

Reviewer 3 Report

This paper asks interesting questions regarding the roles of learning environment, phonological and morphological processing abilities, and cross-linguistic differences among young Urdu-English bilinguals in Pakistan and Canada. The topic is certainly of interest to scholars working in the fields of bilingual literacy, cross-linguistic influence, and second language reading. I believe the paper is publishable; however, there are some important gaps in the paper that need to be addressed:

1.) 1.4.2. Theories of cross-linguistic relations (p. 4). The authors state: "Research findings are inconclusive in terms of the relations among reading skills across languages with similar or different scripts (Gottardo, Chen and Huo 2021; Kim 2013; Schwartz, Leikin and Share 2005; SaieghHaddad and Geva 2008). Therefore, different languages and scripts may require different or modified models of reading to explain developmental pathways and proficiencies (Nag and Snowling 2013). Learners of languages with different scripts and different linguistic typologies provide important contrasts in the search for language-specific and universal "criteria for reading (Share and Daniels 2014)."  Inconclusive in what ways? Please be specific. Also, please be more specific when saying that "different languages and scripts may require different or modified models of reading to explain developmental pathways"? Is there evidence that the pathways themselves are different? If not, what theoretical reasons might there be for different models of reading for different kinds of languages? 

2. Also on page 4: "The languages in the current study, English and Urdu, are alphabetic languages but are written with different scripts. English is written from left to right using the Roman alphabet. Urdu (L1) is written from right to left in Nastaliq script, a modified Arabic/Persian script, which can be written with or without vowel diacritics". What about the orthography/phonology relationship of Urdu? In English, as the authors mention, the orthography does not consistently map on to the phonology.

3. (page 5)"These studies suggest that language and literacy skills are related to each other and L1 and L2 skills can influence each other (Chang 2013; Jiang 2004; Koda 1996; Melby-Lervåg and Lervåg 2011)." Again, this is vague and not new. Both language and literacy are extremely complex. So what are the specific relationships and influences? 

4. Discussion and conclusion. I understand what the authors are trying to say when they state: "These findings challenge the theories developed using data from L2 learners for whom language learning occurs in an immersion context where learners are acquiring the societal language, as is common in the North American or European context. Specifically, the understanding and applicability of linguistic interdependence should be examined and adapted to other linguistic contexts including foreign language learning." (p. 13) However, it is premature to say that the findings challenge a particular theory given that instruction is very different across the two contexts. The authors hint at this a various points in the discussion. But (as with the other points) they are not specific regarding the mechanisms. How exactly might instruction hinder literacy development? It seems to me that it cannot be determined whether the problem is really with the theory or whether the problem is the difference in instruction between the two contexts.  

Author Response

Reviewer 3: (Thank you so much for your review. My responses to your comments and suggestions are addressed in red).

This paper asks interesting questions regarding the roles of learning environment, phonological and morphological processing abilities, and cross-linguistic differences among young Urdu-English bilinguals in Pakistan and Canada. The topic is certainly of interest to scholars working in the fields of bilingual literacy, cross-linguistic influence, and second language reading. I believe the paper is publishable; however, there are some important gaps in the paper that need to be addressed:

1.) 1.4.2. Theories of cross-linguistic relations (p. 4). The authors state: "Research findings are inconclusive in terms of the relations among reading skills across languages with similar or different scripts (Gottardo, Chen and Huo 2021; Kim 2013; Schwartz, Leikin and Share 2005; Saiegh Haddad and Geva 2008). Therefore, different languages and scripts may require different or modified models of reading to explain developmental pathways and proficiencies (Nag and Snowling 2013). Learners of languages with different scripts and different linguistic typologies provide important contrasts in the search for language-specific and universal "criteria for reading (Share and Daniels 2014)."  Inconclusive in what ways?Please be specific. Also, please be more specific when saying that "different languages and scripts may require different or modified models of reading to explain developmental pathways"? Is there evidence that the pathways themselves are different? If not, what theoretical reasons might there be for different models of reading for different kinds of languages? Addressed, please see p. 4, line 165 - 167.

  1. Also on page 4: "The languages in the current study, English and Urdu, are alphabetic languages but are written with different scripts. English is written from left to right using the Roman alphabet. Urdu (L1) is written from right to left in Nastaliq script, a modified Arabic/Persian script, which can be written with or without vowel diacritics". What about the orthography/phonology relationship of Urdu? In English, as the authors mention, the orthography does not consistently map on to the phonology. Added in text. Please see p. 4, line 183 – 187.
  2. (page 5)"These studies suggest that language and literacy skills are related to each other and L1 and L2 skills can influence each other (Chang 2013; Jiang 2004; Koda 1996; Melby-Lervåg and Lervåg 2011)." Again, this is vague and not new. Both language and literacy are extremely complex. So, what are the specific relationships and influences? language is fixed in the paper, p. 5, line 201 - 220.
  3. Discussion and conclusion. I understand what the authors are trying to say when they state: "These findings challenge the theories developed using data from L2 learners for whom language learning occurs in an immersion context where learners are acquiring the societal language, as is common in the North American or European context. Specifically, the understanding and applicability of linguistic interdependence should be examined and adapted to other linguistic contexts including foreign language learning." (p. 13) However, it is premature to say that the findings challenge a particular theory given that instruction is very different across the two contexts. The authors hint at this a various points in the discussion. But (as with the other points) they are not specific regarding the mechanisms. How exactly might instruction hinder literacy development? It seems to me that it cannot be determined whether the problem is really with the theory or whether the problem is the difference in instruction between the two contexts.  This comment has been addressed in paper at various points, more prominently in discussion. Please see changes in track changes in “Discussion”.

Round 2

Reviewer 1 Report

Several of my original concerns have been addressed here. However, I still have two main concerns.

First, the reliability measures for the group in Pakistan were provided. But the group in Canada still represent a departure from the norms for each of the English standardized measures. It would be important to present the reliability measures for this sample as well and not the reported measures from the manuals. 

Second. I still do not understand the purpose or outcome of the analyses listed as Research question "1. Do language-general mechanisms (phonological awareness and morphological skills) influence the relationships across typologically different orthographies?" I did re-read the description in the text, and I still cannot visualize how the analysis was done and what we learn as a reader that differs from RQ2 and RQ3. It also seems to me that the implications from RQ1 are not discussed in the discussion. If this analysis is not worth discussing, then I am not sure the need to present them in the results. I am sorry if I am being obtuse here, but I am a member of your target audience and I could not find this analysis accessible.   

Finally, I noted that in your comparison of performance across groups that there is a different pattern of results for the morphological measures in both Urdu and English. This pattern should require that overall statements about differences across contexts be qualified by discussing the individual skills.  I also wonder whether these comparisons are their own research question. 

Author Response

Reviewer 1 comments and questions:

Several of my original concerns have been addressed here. However, I still have two main concerns.

First, the reliability measures for the group in Pakistan were provided. But the group in Canada still represent a departure from the norms for each of the English standardized measures. It would be important to present the reliability measures for this sample as well and not the reported measures from the manuals. Cronbach’s alpha for the sample tested in Canada has been calculated and added to the paper for all English measures.

Second. I still do not understand the purpose or outcome of the analyses listed as Research question "1. Do language-general mechanisms (phonological awareness and morphological skills) influence the relationships across typologically different orthographies?" I did re-read the description in the text, and I still cannot visualize how the analysis was done and what we learn as a reader that differs from RQ2 and RQ3. It also seems to me that the implications from RQ1 are not discussed in the discussion. If this analysis is not worth discussing, then I am not sure the need to present them in the results. I am sorry if I am being obtuse here, but I am a member of your target audience and I could not find this analysis accessible.  Research questions have been reworded and shifted around in the analyses to address this concern.

Finally, I noted that in your comparison of performance across groups that there is a different pattern of results for the morphological measures in both Urdu and English. This pattern should require that overall statements about differences across contexts be qualified by discussing the individual skills.  I also wonder whether these comparisons are their own research question. Addressed. Please see pg. 12, line 583 – 585.

Reviewer 2 Report

I still think you could better explain the exploratory nature of RQ3 and qualify your findings related to RQ3 a bit more.  As I stated in the first round, this comparison is troublesome, and I'm not sure why it wasn't addressed a bit more.  

Author Response

Reviewer 2 comments and questions:

I still think you could better explain the exploratory nature of RQ3 and qualify your findings related to RQ3 a bit more.  As I stated in the first round, this comparison is troublesome, and I'm not sure why it wasn't addressed a bit more.  This question and the relevant analysis has been explained (reworded) in the current submission.

Round 3

Reviewer 1 Report

The authors have adequately addressed my concerns.